# Antibiotic Prescribing Decisions for Upper Respiratory Tract Infections Among Primary Healthcare Physicians in China: A Mixed-Methods Approach Based on the Theory of Planned Behavior

**DOI:** 10.3390/antibiotics13111104

**Published:** 2024-11-20

**Authors:** Muhtar Kadirhaz, Yushan Zhang, Nan Zhao, Iltaf Hussain, Sen Xu, Miaomiao Xu, Chengzhou Tang, Wei Zhao, Yi Dong, Yu Fang, Jie Chang

**Affiliations:** 1Department of Pharmacy Administration, School of Pharmacy, Xi’an Jiaotong University, Xi’an 710061, China; muhtar@stu.xjtu.edu.cn (M.K.); zys20170521@stu.xjtu.edu.cn (Y.Z.); a528434146@stu.xjtu.edu.cn (N.Z.); iltafhussain@stu.xjtu.edu.cn (I.H.); marsxs@stu.xjtu.edu.cn (S.X.); mm_xu@stu.xjtu.edu.cn (M.X.); 3123315106tcz@stu.xjtu.edu.cn (C.T.); 3124315110@stu.xjtu.edu.cn (W.Z.); dongyi@stu.xjtu.edu.cn (Y.D.); 2Center for Drug Safety and Policy Research, Xi’an Jiaotong University, Xi’an 710061, China; 3Shaanxi Center for Health Reform and Development Research, Xi’an 710061, China; 4Research Institute for Drug Safety and Monitoring, Institute of Pharmaceutical Science and Technology, Western China Science and Technology Innovation Harbor, Xi’an 710115, China

**Keywords:** primary healthcare, antibiotic use, upper respiratory tract infections

## Abstract

**Objectives:** In China, primary healthcare (PHC) facilities have high antibiotic prescribing rates for upper respiratory tract infections (URTIs), which are primarily viral and self-limited. This study aimed to identify the main factors influencing PHC physicians’ antibiotic decisions for URITs based on the theory of planned behavior. **Methods:** A convergent mixed-methods study was conducted at 30 PHC facilities across Shaanxi Province, China. A total of 108 PHC physicians completed a five-point Likert Scale questionnaire focused on behavioral components of antibiotic prescribing, including attitudes, subjective norms, perceived behavioral control, belief in past experiences, and prescribing intentions. Twenty-two physicians participated in semi-structured interviews. **Results:** Respondents had a good awareness of AMR (Mean = 4.49) and a weak belief regarding the benefit of antibiotics (Mean = 2.34). The mean score for subjective norms was 3.36, and respondents had good control over their prescribing behavior (Mean = 4.00). A reliance on past prescribing experiences was observed (Mean = 3.34), and physicians’ antibiotic prescribing intention was 3.40 on average. Multiple linear regression revealed that physicians showing a more favorable attitude towards antibiotics (*p* = 0.042) and relying more on their past experiences (*p* = 0.039) had a higher antibiotic prescribing intention. Qualitative interviews indicated that most physicians would consider prescribing antibiotics when facing diagnostic uncertainty. Low utilization of diagnostic tests, limited effectiveness of training programs, inadequate knowledge of guidelines, and lack of feedback on antibiotic prescriptions all contributed to antibiotic overprescribing. **Conclusions:** PHC physicians in China demonstrated strong intentions to prescribe antibiotics for URTIs when facing diagnostic uncertainty. Beliefs about antibiotics and previous prescribing behavior were significantly linked to prescribing intentions. Multifaceted interventions that focus on facilitating diagnostic tests, improving the quality of training, effectively implementing clinical guidelines, and providing practical feedback on antibiotic prescriptions may help reduce antibiotic overprescribing in China’s PHC facilities.

## 1. Introduction

Antibiotic overprescription is a significant driver of antimicrobial resistance (AMR), which has emerged as a critical global public health issue [1]. AMR leads to prolonged illnesses, higher mortality rates, and increased treatment costs [2]. Antibiotics are widely prescribed in primary healthcare (PHC) facilities, with upper respiratory tract infections (URTIs) being the most common reason for these prescriptions [3,4]. Etiological studies indicate that the majority of URTIs are viral in origin, with only less than 20% of cases attributed to bacterial infections [5]. The use of antibiotics offers limited benefits in the treatment of most URTI cases. According to the disease-specific quality indicators proposed by the European Surveillance of Antimicrobial Consumption Project (ESAC), the percentage of outpatient URTI visits prescribed antibiotics should not exceed 20% [6]. However, a recent nationwide survey revealed that 55.1% of URTI visits were prescribed antibiotics at PHC facilities in China [7]. These rates were higher than those reported in Australia (12%), Finland (8.8%), and Sweden (4.5%) [8,9,10]. Furthermore, broad-spectrum antibiotics, especially second- and third-generation cephalosporins, were the most commonly prescribed antibiotics for URTIs at PHC facilities in China [7]. Research has shown that overprescribing antibiotics in primary healthcare facilities increases the prevalence of antibiotic-resistant organisms [11]. Data from the China Antimicrobial Surveillance Network (CHINET) revealed that in 2022, the resistance rates of *Klebsiella pneumoniae* isolates to ceftriaxone and cefepime were 42.7% and 35.9%, respectively [12]. For *Streptococcus pneumoniae*, the resistance rates to clindamycin and erythromycin remained constantly high (>90%) from 2015 to 2021 [13]. Therefore, overprescribing of antibiotics for URTIs is a matter of serious concern in PHC facilities in China.

To improve the quality of antibiotic prescribing in PHC facilities, several interventions have been proposed in the past years [14]. The successful implementation of these interventions in real-world settings requires a thorough understanding of PHC physicians’ prescribing behavior. Previous studies have identified various factors influencing antibiotic prescribing practices, including physician-related factors such as knowledge, attitude, and workload, as well as relationships with patients and cultural factors [15]. Due to the complexity of the healthcare environment, these factors can have varying influences across different settings. For instance, while Fletcher-Lartey et al. found that PHC physicians’ attitude to AMR significantly impacted antibiotic prescribing for URTI patients in Australia [16], a study from the UK suggested that physicians’ experience and confidence in clinical decision-making are key factors in determining antibiotic prescribing practices [17]. Therefore, understanding physicians’ antibiotic prescribing behavior within various healthcare systems and socio-cultural contexts is essential for the successful implementation of interventions designed to address antibiotic overprescription.

Previous studies from China have identified several factors contributing to antibiotic prescribing in PHC facilities, including physicians’ professional knowledge, attitude, diagnostic capabilities, patient pressure, and financial incentives [18,19,20]. Only a few of these studies focused on URTIs [21]. Additionally, the subjective opinions held by PHC physicians have often been overlooked. To date, qualitative data investigating PHC physicians’ prescribing practices for URTIs are scarce in China [22]. These gaps hinder a comprehensive understanding of the factors influencing antibiotic prescription for URTIs in China’s PHC facilities. Therefore, we conducted a mixed-methods study to investigate how PHC physicians make decisions regarding antibiotic prescription for URTIs, and to identify key factors that influence this decision-making process. We believe our findings can provide valuable evidence for designing and implementing effective interventions to reduce antibiotic overprescription in PHC settings.

## 2. Materials and Methods

### 2.1. Study Design

A convergent mixed-methods design was used. Quantitative and qualitative data were collected in parallel, analyzed separately, and then merged to provide a comprehensive understanding of physicians’ antibiotic prescribing behavior regarding URITs [23]. Quantitative data was comprised of PHC physicians’ responses to a questionnaire survey. Qualitative data was collected from semi-structured interviews with PHC physicians.

The theory of planned behavior (TPB) assumes that behavior intentions are the best predictor of the actual behavior. These intentions are influenced by attitudes toward the behavior, subjective norms, and perceived behavioral control [24]. In addition, previous studies have shown that past experience is significantly related to behavior intention and actual behavior and has been added as an additional predictor in the TPB model [25,26]. Therefore, we adapted an extended model of the TPB, using belief in experience to measure the influence of past behavior on physicians’ prescribing intentions (Figure 1).

### 2.2. Participants

Our study was conducted in Shaanxi Province, China. In 2020, the gross domestic product (GDP) of Shaanxi Province was CNY 2618.2 billion (USD 379.5 billion), ranking 15th among the 31 provinces in the country [27]. We purposely selected three out of ten prefecture-level cities (Xi’an, Yan’an, and Shangluo) based on their geographic locations and economic development levels (Appendix A). A convenience sampling strategy was used to recruit participants. In each selected prefecture-level city, community healthcare centers (CHCs) in urban areas and township healthcare centers (THCs) in rural areas were visited by investigators from February 2021 to July 2021. The purpose and procedure of the study were thoroughly explained to facility directors and physicians. Physicians who prescribed antibiotics independently at the participating facilities were included in our study. Participation was voluntary and anonymous. Details of the recruitment process are provided in Appendix A.

### 2.3. Data Collection

#### 2.3.1. Quantitative Data Collection

Quantitative data were collected through a self-administered questionnaire. The questionnaire design was based on previous studies and the guidance of the TPB [28,29,30]. Questions were designed using a five-point Likert Scale to measure different behavioral constructs:

Attitudes were measured by using 7 items to assess physicians’ perceptions of the consequences of prescribing antibiotics. Two subscales were developed, including awareness of AMR (3 items) and beliefs about the effectiveness of antibiotic treatment for URTIs (4 items). Respondents’ answers were rated from 1 “strongly disagree” to 5 “strongly agree”. A higher score indicated a better awareness of AMR and a stronger belief in antibiotic effectiveness.

Subjective norms were measured using 4 items, which assessed perceived social pressure from patients and colleagues. Respondents’ answers were rated from 1 “strongly disagree” to 5 “strongly agree”. A higher score indicated a higher perceived social pressure.

Perceived behavioral control was measured by using 5 items to estimate the difficulty or ease of executing the prescribing behavior. Respondents’ answers were rated from 1 “strongly agree” to 5 “strongly disagree”, with a higher score indicating a higher behavioral control.

Belief in experience was measured by using 5 items to evaluate physicians’ perceptions of their past prescribing behavior. Respondents’ answers were rated from 1 “strongly disagree” to 5 “strongly agree”. A higher score indicated more reliance on prior prescribing experience.

Prescribing intentions were measured by 6 items, which assessed the degree to which a physician is willing to prescribe antibiotics for URTI patients. Respondents’ answers were rated from 1 “never” to 5 “always”, with a higher score indicating a higher prescribing intention.

Physicians’ demographic characteristics included in the study were age, gender, education level, years of practice, professional title, training experience, location, and the type of facility. Data regarding the availability of diagnostic tests such as routine blood tests and C-reactive protein tests in participating facilities were also collected.

The questionnaire was submitted in a pilot test to ten PHC physicians to check the face validity and clarity of the questions. Subsequent changes were made upon receiving their feedback. Eligible physicians were approached and invited to participate in the survey. All questionnaires were completed onsite without referring to external sources of information. Returned questionnaires were examined by investigators, with missing items being rechecked and refilled immediately on-site. The internal consistency of the questionnaire was tested in the final survey (n = 108). Acceptable internal consistency was indicated by Cronbach’s alpha (0.545–0.841) for the measured constructs.

#### 2.3.2. Qualitative Data Collection

A semi-structured interview guide was developed based on a thorough literature review [22,30,31,32] and the guide of the TPB. The interview guide included open questions and follow-up probes to elucidate physicians’ attitudes about AMR and antibiotic use, physicians’ antibiotic practices for URTIs, factors affecting prescribing decisions, and their sources of information. The questions were reviewed by a team of two infectious disease physicians. The wording and the relevance of the questions were checked. A pilot study was conducted on two potential participants (not included in the final data analysis) to check their understanding of the questions. Minor modifications were made according to their feedback. The interview guide can be found in Appendix A. At least one eligible physician was invited and interviewed in each facility. The interviews were conducted in a quiet room at the participating facility. All interviews were audio recorded with the participants’ consent. The principle of data saturation was used.

### 2.4. Data Analysis

#### 2.4.1. Quantitative Data Analysis

Demographic characteristics of participating physicians were summarized using descriptive statistics. The frequency and percentage of each point class of the Likert Scale were calculated, as well as the mean score and standard deviation for each behavioral construct measurement, with a mean score of 3 representing a neutral response. In addition, we used multiple linear regression to investigate possible relationships between physician’s prescribing intention and its antecedents (attitudes, subjective norms, perceived behavioral control, and belief in experiences) [33], with physicians’ gender, age, educational level, professional title, years of working and facility type included as covariates. Model checking was undertaken using residual plots. The coefficient of determination (adjusted R^2^) was used to assess the proportion of the variability in intention explained by the model. The significance level was set to 5%. All quantitative analysis was conducted using Stata 16 (Stata Corp, College Station, TX, USA).

#### 2.4.2. Qualitative Data Analysis

Thematic analysis of the data was undertaken following the guidelines outlined by Braun and Clarke [34]. The analysis was initially conducted in Chinese. All recording interviews were transcribed verbatim. First, the transcripts were checked against the audio recordings for accuracy, and then the transcripts were read several times for familiarization. Second, the viewpoints and sentences associated with the research objectives were coded. Next, the initial codes were categorized into relevant themes and revised iteratively through ongoing discussion until all researchers reached a consensus. Finally, the results were compiled and summarized, with themes and relevant quotes translated into English. The analysis was performed using NVivo 12 software (QSR International, Melbourne, Australia).

#### 2.4.3. Mixed-Methods Integration

Informed by a convergent approach, the results from the quantitative study were compared with the results from the qualitative study [35]. The study team discussed and considered how quantitative and qualitative findings confirm, contradict, and expand understanding of PHC physicians’ antibiotic prescribing behavior for URTIs.

### 2.5. Ethical Considerations

Ethics approval was obtained from the Biomedical Ethics Committee of Xi’an Jiaotong University (No. 2019-1241). An information letter and consent form were provided to all participants, with a comprehensive description of their rights, including voluntary participation and their right to withdraw consent at any time during the study.

## 3. Results

A total of 119 eligible physicians at 30 PHC facilities across Shaanxi Province were approached, and 108 (90.8%) completed the questionnaire. Of those who filled out the questionnaire, 22 participated in semi-structured interviews. All PHC facilities provide routine blood tests, and the C-reactive protein test was available in 36.7% of facilities. Demographic characteristics of questionnaire respondents and interview participants are listed in Table 1.

### 3.1. Quantitative Results

Figure 2 shows the proportion of physicians that agreed or strongly agreed with the statements regarding different behavioral components (always or often for prescribing intentions). The mean measurement scores of survey respondents for each behavioral component are listed in Table 2. Detailed results of survey responses are in Appendix A.

The survey showed that the majority of respondents had a good awareness of AMR (Mean = 4.49, SD = 0.50). Over ninety-five percent of the respondents (104/108) agreed that AMR is a major public health issue in China. All physicians believed that overprescribing antibiotics could lead to AMR. Most physicians (66.7%) acknowledged that antibiotic overprescription is common in PHC facilities in China. Our respondents showed a weak belief in the effectiveness of prescribing antibiotics for URTIs (Mean = 2.34, SD = 0.80). Only a small percentage of respondents believed that antibiotic treatment for URTIs could relieve symptoms (6.5%), lower relapse rates (11.1%), and prevent exacerbation (8.3%) or severe complications (11.1%).

An inclination to rely on prior antibiotic prescribing experience was observed (Mean = 3.34, SD = 0.71). Nearly 60% (57.4%) of our respondents believed that prior experience was important for their daily antibiotic prescribing practices, especially when diagnostic tests were not available (64.8%). Nearly 40% (37.1%) of physicians agreed that most of their prescribing decisions were according to their prior antibiotic prescribing behavior. Many physicians stated that prior prescribing experience could help them deal with diagnostic uncertainty (49.1%) and improve consultation efficiency (42.6%).

Social pressure (subjective norms) from patients and colleagues was evident (Mean = 3.36, SD = 0.65). Over 60% of respondents acknowledged that they had perceived pressure for prescribing antibiotics from patients (63.9%) or parents (60.2%). The prescribing behavior of senior physicians had a greater impact on respondents’ antibiotic prescribing practices than that of peer physicians (50.9% vs. 25.9%). Our survey respondents believed that they had good control over their antibiotic-prescribing behavior (Mean = 4.00, SD = 0.56). Only 15.7% of physicians stated that the decision to prescribe antibiotics is not entirely up to themselves. A very small percentage of physicians agreed that they would prescribe antibiotics if strongly requested by patients (6.5%) or parents (4.6%). Similarly, our respondents rarely prescribe antibiotics when they do not have time to explain their decisions (1.9%) or assess patients’ conditions (3.7%).

The mean score of prescribing intention was 3.40 (SD = 0.86). Nearly 90% (87.1%) of the respondents always or often prescribe antibiotics to URTI patients with suppurative tonsillitis, followed by purulent sputum (59.3%), cough (52.8%), nasal discharge (46.3%), fever over 38.5 °C (36.1%), and sore throat (19.4%).

Based on the regression analysis, the behavioral intention antecedents explained approximately one-fifth of the variability in physician’s prescribing intention (adjusted R^2^ = 0.21). An attitude in favor of antibiotic effectiveness (β = 0.251, *p* = 0.042) and a greater reliance on prior antibiotic prescribing experience (β = 0.242, *p* = 0.039) were statistically related to a higher intention to prescribe antibiotics (Table 2).

### 3.2. Qualitative Results

Upon analyzing the interview scripts, four themes were generated consisting of ten sub-themes, as summarized in Table 3.

#### 3.2.1. Theme 1: Treatment of URTIs in PHC Facilities

Sub-theme 1-1: Antibiotic prescribing practices for URTIs

All participants stated that acute URTIs accounted for a large proportion of the diseases managed in their everyday practice. Around half of the respondents (10 out of 22) stated that antibiotics were not necessary for URTIs and that only symptomatic treatment was required. However, most physicians admitted that they would consider prescribing antibiotics if they were uncertain whether bacteria or viruses were causing the conditions. The most common conditions that would incur antibiotic prescription include fever > 38.5 °C (14/22), ranked first, followed by yellow phlegm (11/22), pus on the tonsils (10/22), swollen throat (8/22), and green nasal discharge (7/22). Some participants said they would prescribe antibiotics if the symptoms lasted over three or four days.


*“Green nasal discharge, sore throat, and yellow phlegm mean you might suffer a bacterial infection. Especially when your body temperature is above 38.5 °C, I will consider using antibiotics.”*



*“…if a patient has a cough or fever for more than three or four days, it is probably a bacterial infection, and I will prescribe antibiotics in these situations.”*


Sub-theme 1-2: Low utilization of diagnostic tests

All respondents reported having clinical laboratory departments in their facilities; however, many participants mentioned that diagnostic tests were not fully utilized. Some physicians stated that understaffing is the primary reason for the low utilization of laboratory tests. Respondents also stated that patients often decline to take diagnostic tests due to inability to afford the tests or lack of time. Additionally, some physicians in rural areas see very few patients, making it uneconomical to run laboratory tests for only one or two samples.


*“We only have one laboratory staff member; if he is off-duty, we cannot do any tests, and I have to prescribe all by my own experience.”*



*“Sometimes we suggest patients to get blood tests, but they don’t listen. They don’t want to pay extra money. They think it is not necessary for a cold.”*



*“Our lab department sometimes declines to do the test because they feel it is not worth starting up the machine for just one sample.”*


Sub-theme 1-3: Confidence in previous prescribing experience

The participants demonstrated high confidence in their previous antibiotic-prescribing experiences. Most believed they could make rational decisions based on their prior prescribing behavior. Some physicians would rather rely on personal experiences than diagnostic test results. Some also said that when the initial symptomatic treatment was ineffective, their habitual approach was to change to antibiotics without further evidence.


*“I have been working for nearly forty years. The symptoms are obvious, and there is no need for laboratory tests.”*



*“I prescribe antibiotics mainly according to patients’ symptoms and my own experience. I am pretty sure when antibiotics are necessary.”*


#### 3.2.2. Theme 2: Attitudes

Sub-theme 2-1: Awareness of AMR

Almost all of our participants said they had heard of the term “AMR”, and acknowledged that it is a serious public health issue in China. However, most respondents lacked awareness regarding AMR’s accurate definition and mechanism. Most physicians mistakenly perceived AMR as the human body being resistant to antibiotics. Furthermore, although nearly all physicians acknowledged that antibiotic overprescription contributes to the development of AMR, none of the participants took it into consideration while prescribing antibiotics. All participants agreed that antibiotic overuse is common in PHC facilities in China. Nevertheless, most physicians believed that the primary responsibility lies with village clinics and private facilities.


*“…if you take antibiotics frequently, your body will become less sensitive to the medicine.”*



*“…village clinics prescribe far more antibiotics than us, irrespective of the severity of URTIs, sometimes they prescribe two or three kinds of antibiotics for patients.”*


Sub-theme 2-2: Perceptions on the role of antibiotics

A majority of the respondents (16/22) stated that antibiotics provide little benefit for URTI patients. Only a few participants (5/22) stated that they had prescribed antibiotics for the prevention of possible complications. However, most physicians believed that broad-spectrum antibiotics give better clinical outcomes than narrow-spectrum ones. Additionally, most participants agreed that when they were uncertain about the etiology, or without diagnostic test results like leukocyte counts, broad-spectrum antibiotics could avoid misdiagnosis by covering a multitude of pathogens. A significant proportion of the respondents (18/22) stated that their first choice of antibiotics would be broad-spectrum antibiotics. Cephalosporins (17/22) and amoxicillin (13/22) were the most commonly used antibiotics.


*“…we can’t do susceptibility tests to identify the pathogen, so it’s better to use broad-spectrum antibiotics to cover all kinds of possible bacteria.”*



*“…my experience tells me when you are not sure what kind of infection it is, you’d better use broad-spectrum antibiotics, they are more effective.”*



*“I prefer broad-spectrum antibiotics. Mostly we use amoxicillin, clarithromycin, and cephalosporins such as cefotaxime, ceftriaxone.”*


#### 3.2.3. Theme 3: Sources of Information

Sub-theme 3-1: Limited effectiveness of training programs

Most of our participants had attended continuous medical training in the past twelve months. The most common type of training was lectures hosted by regional health bureaus. However, physicians complained that the current training did not meet their needs in PHC settings. Over half of the participants (13/22) expressed a need for knowledge on optimal selection, dosage, and duration of antibiotic treatment. Some physicians stated that the current continuing training courses cover many aspects, but antibiotics only account for a small part.


*“I recently took part in some training sessions where experts from big hospitals gave lectures. However, I didn’t find the courses very useful as they didn’t seem to relate to our daily practices.”*



*“I do like to take part in some training programs. I’d like to learn more about the selection, dosage and usage of antibiotics.”*


Sub-theme 3-2: Inadequate knowledge of clinical guidelines

None of our participants could specify any available guidelines on the use of antibiotics in their facilities during our visits. Most physicians were unaware of where to obtain clinical guidelines. Some participants said they had learned about the national guidelines for antibiotic use in clinical practice. However, few of them could identify any regulations on the clinical application of antibiotics. Only three physicians were fully aware of the requirements for the antibiotic prescribing rate in outpatient settings. Nearly all physicians acknowledged that they knew little about any clinical guidelines for acute URTIs.


*“There are guidelines about hypertension, but I never heard about any guidelines for acute URTIs.”*



*“No one asks us to learn any guidelines on antibiotics. I don’t know where to get these guidelines, maybe from websites, I guess…”*


Sub-theme 3-3: Limited feedback on antibiotic prescriptions

Over half of the respondents said outpatient antibiotic prescribing rates were tracked and reported in their facilities. Nevertheless, antibiotic use for specific conditions, such as acute URTIs, was not calculated. All participants were unaware of their antibiotic prescribing rate for acute URTI cases. Most respondents (16/22) reported that prescriptions were reviewed by senior physicians or higher authorities monthly or quarterly. However, as reported by our respondents, prescription reviews conducted in their facilities were merely a formality. The rationality of prescribing antibiotics was not reviewed due to the low quality of medical records. Symptoms and laboratory test results were not recorded in health information systems in most facilities, making it difficult to evaluate whether the antibiotics were prescribed according to guidelines. None of our respondents had received any feedback on the appropriateness of their antibiotic prescriptions.


*“…we don’t know how many antibiotics I prescribed to URTI patients. No one ever told us about it…”*



*“It (prescription review and feedback) doesn’t provide much help. They only check if your prescription matches your diagnosis. For example, you cannot prescribe antibiotics for hypertension. However, for acute URTIs, it’s difficult to determine if your prescription is appropriate. Only final diagnoses are recorded, and detailed information like symptoms or diagnostic tests are not documented, so it’s hard to tell if antibiotics are necessary.”*


#### 3.2.4. Theme 4: Social Pressure

Sub-theme 4-1: Patient’s demand for antibiotics

All the interview participants stated that patients often requested antibiotics because they believed antibiotics were a quick fix for their illness. The demand for antibiotics was observed more frequently among older patients than younger patients, and rural residents were perceived as more likely to request antibiotics. Additionally, some patients requested antibiotics for their family members or kept them on hand “just in case.” Most participants (14 out of 22) stated that they would not prescribe antibiotics if they felt it was unnecessary, even if patients insisted. Some physicians mentioned that diagnostic test results helped explain their prescription decisions.


*“Some patients, especially elderly ones, frequently request antibiotics, but I will not prescribe them if the medication is unnecessary, even if they insist.”*



*“Sometimes, I show patients their blood test results and explain that antibiotics are unnecessary because their leukocyte count is low.”*


Sub-theme 4-2: Peer influence

Nearly all participants mentioned that their colleagues had prescribed antibiotics to patients without clear clinical indications. Nevertheless, most of our respondents indicated that the prescribing behavior of other physicians would not influence them. Most participants rarely change a peer’s prescription, even if there is no clear evidence for antibiotic treatment. However, some physicians mentioned discussing it with their colleagues after the consultation. Only a few respondents (3/22) said they would seek advice from senior physicians.


*“I know some physicians always prescribe antibiotics; this is their habit. But I have my own judgement, I won’t prescribe like them”*



*“Sometimes I see my colleagues prescribing antibiotics unnecessarily. But I won’t say it in front of the patient. I may talk about it later.”*


### 3.3. Mixed-Methods Integration

A comparison of the qualitative and quantitative findings is provided in Table 4. The survey responses showed a strong inclination towards prescribing antibiotics for URTIs. In the qualitative data, physicians elaborated on this, stating that uncertainty in diagnosis was the primary reason for prescribing antibiotics. The survey also indicated that PHC physicians were aware that overprescribing antibiotics is a significant cause of AMR. The interview results expanded on this notion, where a reluctance to take responsibility was described. Physicians believed that village clinics and private facilities were more responsible for antibiotic misuse and the development of AMR. Furthermore, the survey responses indicated that physicians believed antibiotics had limited benefits for URTI patients. The interviews confirmed and expanded on this, with physicians stating that broad-spectrum antibiotics can minimize diagnostic uncertainty by covering a multitude of pathogens.

The quantitative results revealed an inclination to rely on prior antibiotic prescribing experience among PHC physicians. This finding is consistent with the qualitative results, where participants stated that past prescribing experience is essential in their antibiotic decisions. In addition, most survey respondents reported that they had good control over their prescribing behavior. This finding is also found in the interviews, where most physicians refused to fulfill patients’ requests for antibiotics if unnecessary.

As indicated from the survey responses, around eighty percent of physicians received at least one training session on antibiotics in the last year. However, most interviews suggested that these trainings provided limited help, and physicians demonstrated poor knowledge regarding clinical guidelines on antibiotics or acute URTIs. Furthermore, although all facilities had a laboratory department and routine blood tests were available, the qualitative data indicated that diagnostic tests were not fully utilized due to understaffing or patient resistance.

## 4. Discussion

In this mixed-methods study, we investigated the factors that influence physicians’ decisions to prescribe antibiotics for URTIs in PHC facilities in China. To gain a comprehensive understanding of this process, we used an extended TPB model as the theoretical framework. The insights gained from this study can guide targeted interventions aimed at reducing antibiotic overuse and combating the escalating threat of AMR.

We found that physicians with favorable attitudes toward antibiotic effectiveness had a higher prescribing intention. Our participants acknowledged that the effectiveness of antibiotic treatment for URTIs is marginal. However, when facing diagnostic uncertainty, most of our participants believed that antibiotics, especially broad-spectrum antibiotics, can minimize the risk of misdiagnosis. Diagnostic uncertainty has been identified as a key driver for the overprescribing of antibiotics [36]. Previous research has shown that diagnostic tests can reduce clinical uncertainty and improve antibiotic prescribing practices [37]. Nowadays, most PHC facilities in China can provide diagnostic tests, such as routine blood tests, to help physicians identify possible bacterial infections [38]. However, our findings revealed that these tests were not fully utilized. Some physicians rely on their prior experience rather than diagnostic test results. Moreover, patient resistance to diagnostic test-taking is common. A previous study showed that a majority of patients in the U.S. were willing to have a blood test to guide antibiotic use for respiratory tract infections [39]. Further interventions are needed to improve patient’s willingness to take diagnostic tests in China’s PHC facilities.

Our results revealed that physicians who relied more on their past experience tended to prescribe more antibiotics for URTI patients. However, in contrast, some previous studies showed that experienced physicians prescribe fewer antibiotics than those who have less working experience [40,41]. Physicians’ experience could influence the prescribing of antibiotics in either direction [42]. A study from the Netherlands revealed that for physicians with relatively little professional knowledge, the longer they had practiced, the more frequently they prescribed antibiotics [43]. In China, PHC physicians demonstrated insufficient knowledge of antibiotic use and poor diagnostic ability [18,19]. It is difficult for PHC physicians in China to make rational prescribing decisions solely based on their personal clinical experience, especially when facing clinical uncertainty [22]. Participants in our study showed a reliance on their past prescribing practices. A similar phenomenon has also been described in a previous study conducted in the U.S., where difficulty in changing physicians’ old practices was a main barrier to appropriate antibiotic prescribing [44]. It has been proved that prescription review and feedback can facilitate physicians to reflect on their past prescribing behavior [45]. Nevertheless, due to the low quality of medical records in China’s PHC facilities, practical feedback on antibiotic prescriptions has yet to be achieved [46]. Physicians were unaware of the appropriateness of their antibiotic prescriptions, which made them more dependent on their past experiences.

Our findings revealed that current training sessions had limited effects on improving antibiotic prescribing in China’s PHC facilities. Although continuing medical education (CME) is a mandatory requirement in most provinces in China [47], our participants found that CME courses provided limited information and were mostly ineffective. One possible explanation could be that CME courses were primarily provided in lecture format only, and focused on theoretical concepts rather than real practice settings [48]. Additionally, we found that physicians were unaware of the clinical guidelines for the diagnosis and treatment of acute URTIs. The Chinese Medical Association issued guidelines for primary care of acute URTIs in 2018 [49]. However, there were no regulations or policies regarding the compulsory implementation of these guidelines in PHC facilities. Targeted interventions are needed to improve PHC physicians’ awareness and adherence to clinical guidelines.

Similar to previous studies [50,51], our participants were concerned about AMR and acknowledged that overprescribing antibiotics contributes to this issue. However, an avoidance of responsibility was observed. Most physicians may not feel responsible for the development of AMR. Furthermore, although participants in our study perceived high patient expectations for antibiotics, they had a high level of behavioral control toward antibiotic prescribing. This finding is similar to previous studies from China and Saudi Arabia [28,52].

Our study has several policy implications for regulating antibiotic use in PHC facilities. Firstly, practical feedback on antibiotic prescribing should be conducted regularly. To achieve this, the quality of medical records should be improved, with patients’ symptoms, history, and diagnostic test results being completely documented. Disease-specific quality indicators should be developed to ensure targeted feedback on antibiotic prescriptions. Computer-based feedback interventions have gained widespread interest nowadays. A previous study in China showed that a computer network-based feedback program can significantly reduce the antibiotic prescription rates of PHC physicians [53]. Secondly, the use of routine blood tests, recommended by the current Chinese guideline for the diagnosis of URTIs [49], should be strengthened. Incorporating the idea of shared decision-making may help improve patients’ willingness to take diagnostic tests. Careful explanations of the uncertainty in diagnosis and the indispensability of the diagnostic tests might help clear out patients’ concerns over taking diagnostic tests [54]. Moreover, to encourage PHC physicians to order diagnostic tests, evidence should be provided regarding the effect of diagnostic tests on patient outcomes and antibiotic prescriptions. Thirdly, training sessions should be tailored for PHC physicians and contain feasible recommendations. Local administrations should enhance the quality of CME for PHC physicians, targeting the optimal selection and usage of antibiotics in PHC settings. Efforts should be made to improve PHC physicians’ awareness and adherence to clinical guidelines. It is essential to establish an open-access national guideline database, which could help physicians obtain clinical guidelines [55]. Lastly, vaccines, such as pneumococcal and influenza vaccines, can help reduce the incidence of respiratory tract infections and, consequently, the use of antibiotics [56]. However, previous studies in China have shown that the coverage rates for these vaccines are low. A nationwide survey conducted in 2019 revealed that in China, the coverage rates for the 1-dose and 3-dose PCV13 (13-valent pneumococcal conjugate vaccine) among children aged 6 to 59 months were only 7.7% and 5.1%, respectively [57]. Additionally, influenza vaccination coverage during the 2020–2021 and 2021–2022 epidemic seasons was only 3.16% and 2.47%, respectively, in China [58]. Therefore, alongside interventions aimed at changing physician behavior, efforts should also focus on increasing vaccination coverage against respiratory tract infections.

The use of a mixed-methods approach strengthened the validity of the study, providing a more comprehensive understanding of factors influencing antibiotic prescribing [59]. However, the limitations of this study should be acknowledged. Firstly, our study relied on physician recall and self-reported practices. Participants may have tended to provide answers they believed the researchers wanted to hear. To reduce information bias, all participants were told that the study was not to identify inappropriate prescribing but to understand physicians’ prescribing processes. Secondly, the study had a small number of respondents. However, our participants were recruited from different regions of the province with different economic development levels (high, medium, and low). Our participants’ characteristics, such as age, education level, and professional title, matched previous studies conducted in China. Therefore, we believe that our sample is representative and can reflect PHC physicians’ prescribing behavior. Thirdly, only physicians from CHCs and THCs were included in the study. While nearly all CHCs and THCs in China are public facilities, almost forty percent of village clinics are privately owned [38]. Views among physicians practicing in village clinics might differ from the findings in this study. Due to the difficulty of data collection, village clinicians were not included in this study. Further studies are needed to explore village clinicians’ antibiotic-prescribing behavior.

## 5. Conclusions

PHC physicians in China demonstrated strong intentions to prescribe antibiotics for URTIs when facing diagnostic uncertainty. Broad-spectrum antibiotics were the preferred initial choice for most physicians. Beliefs about antibiotics and previous prescribing behavior were significantly linked to prescribing intentions. Most PHC physicians relied on their past experiences. Factors such as infrequent use of diagnostic tests, limited effectiveness of current training programs, inadequate knowledge of clinical guidelines, and lack of feedback on antibiotic prescriptions contributed to the reinforcement of misconceptions about antibiotics, leading physicians to rely on their past prescribing experience. Multifaceted interventions that focus on facilitating diagnostic tests, improving the quality of training, effectively implementing clinical guidelines, and providing practical feedback on antibiotic prescriptions may help reduce antibiotic overprescribing in China’s PHC facilities.

## Figures and Tables

**Figure 1 antibiotics-13-01104-f001:**
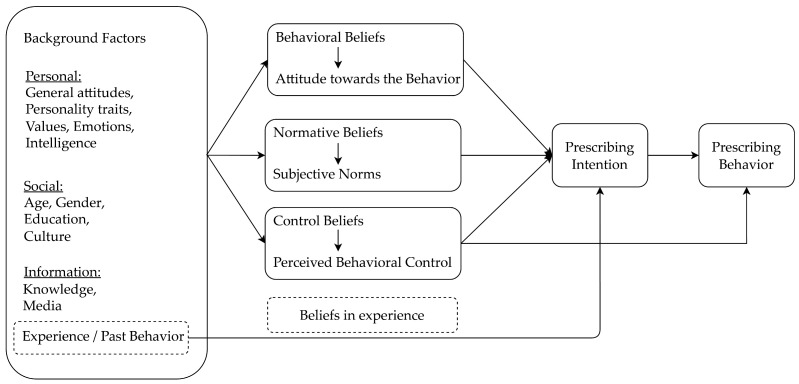
The theoretical framework of an extended model of the theory of planned behavior.

**Figure 2 antibiotics-13-01104-f002:**
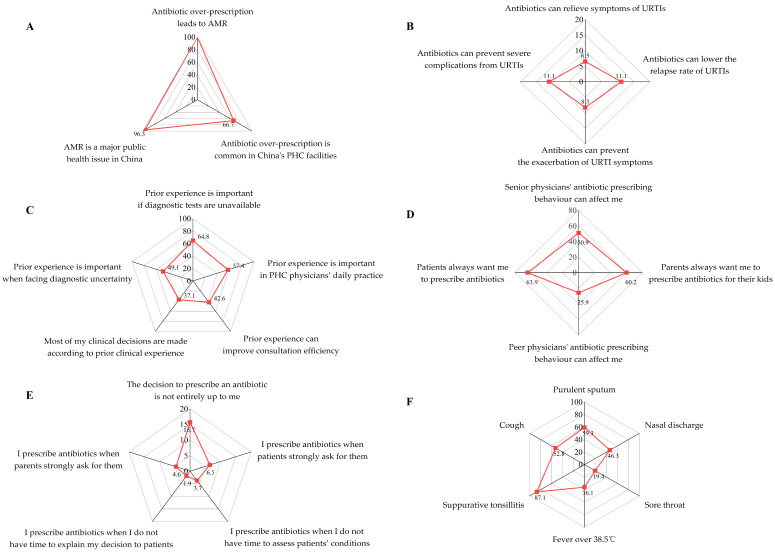
Physicians’ responses for survey items within different behavioral components. (**A**–**E**) Proportion of respondents that strongly agreed or agreed with statements regarding AMR awareness (**A**), belief in antibiotics (**B**), belief in prior prescribing experience (**C**), subjective norms (**D**), and perceived behavioral control I. (**F**) Proportion of respondents that always or often prescribe antibiotics for URTI patients with different symptoms.

**Table 1 antibiotics-13-01104-t001:** Basic characteristics of surveyed physicians and interview participants from 30 primary healthcare (PHC) facilitates in Shaanxi Province, China.

Characteristics	Survey Respondents n (%)	Interview Participants n (%)
Total	108	22
Regions		
Yan’an	22 (20.4)	6 (27.3)
Xi’an	41 (38.0)	6 (27.3)
Shangluo	45 (41.7)	10 (45.4)
Facility type		
CHCs	52 (48.1)	11 (50.0)
THCs	56 (51.9)	11 (50.0)
Age, years (Mean ± SD)	43.5 ± 10.4	43.4 ± 9.8
Gender		
Male	48 (44.4)	11 (50.0)
Female	60 (55.6)	11 (50.0)
Education		
Junior college or below	47 (43.5)	10 (45.5)
Bachelor or above	61 (56.5)	12 (55.5)
Professional title		
Resident	34 (31.5)	9 (40.9)
Attending	53 (49.1)	10 (45.5)
Associate chief	21 (19.4)	3 (13.6)
Years of working (Mean ± SD)	19.4 ± 10.9	19.1 ± 10.2
Training on antibiotics in last 12 months		
None	21 (19.4)	4 (18.2)
Once	35 (32.4)	5 (22.7)
Twice	29 (26.9)	9 (40.9)
More than twice	23 (21.3)	4 (18.2)
Availability of diagnostic tests *		
Routine blood tests	30/30 (100.0)	
C-reactive protein test	11/30 (36.7)	

* Calculated at the facility level. CHCs: community healthcare centers. THCs: township healthcare centers.

**Table 2 antibiotics-13-01104-t002:** Mean scores of measurements and estimated coefficients of multiple linear regression for the relationship between physician’s prescribing intention and its antecedents (attitudes, subjective norms, perceived behavioral control, and belief in experience).

Measurements	Mean ± SD Range (0 to 5)	Coefficients	*p* Value
Attitudes			
Awareness of AMR (3 items)	4.49 ± 0.50	0.117	0.464
Belief in antibiotics (4 items)	2.34 ± 0.80	0.251	0.042
Subjective norms (4 items)	3.36 ± 0.65	0.075	0.564
Perceived behavioral control (5 items)	4.00 ± 0.56	−0.093	0.582
Belief in experience (5 items)	3.34 ± 0.71	0.242	0.039
Prescribing intention (6 items)	3.40 ± 0.86	/	/

SD: standard deviation.

**Table 3 antibiotics-13-01104-t003:** Themes and sub-themes generated from qualitative interviews.

Themes	Sub-Themes
Treatment of URTIs in PHC facilities	Antibiotic prescribing practices for URTIs
	Low utilization of diagnostic tests
	Confidence in previous prescribing experience
Attitudes	Awareness of AMR
	Perceptions on the role of antibiotics
Sources of information	Limited effectiveness of training programs
	Inadequate knowledge of clinical guidelines
	Limited feedback on antibiotic prescriptions
Social pressure	Patient’s demand for antibiotics
	Peer influence

**Table 4 antibiotics-13-01104-t004:** Quantitative and qualitative results merging.

Over-Arching Themes	Quantitative Findings	Qualitative Findings	Meta-Inferences
Prescribing intentions	The mean score of prescribing intention was 3.40, and the intention varies between different symptoms.	Physicians were aware that most URTIs are viral. However, respondents intended to prescribe antibiotics when they were uncertain if the symptoms were caused by bacteria or viruses.	Expanded. When facing diagnostic uncertainty, antibiotics were frequently prescribed for URTI patients.
Awareness of AMR	Physicians had a good awareness of AMR (Mean score = 4.49).	Physicians acknowledged that antibiotic overprescription is a serious problem at the PHC level. However, respondents believed that village clinics and private facilities were more responsible for this issue.	Expanded. Physicians generally concerned about antibiotic overprescription but blamed others for the development of AMR.
Belief in antibiotics	Physicians had a weak belief about the benefits of antibiotic treatment for URTIs (Mean score = 2.34).	Physicians acknowledged that antibiotics cannot relieve URTI symptoms or prevent any complications. However, participants believed that broad-spectrum antibiotics can minimize risk when facing diagnostic uncertainty.	Expanded. Misconceptions of the role of broad-spectrum antibiotics were prevalent among PHC physicians. Broad-spectrum antibiotics were physicians’ first choice.
Belief in past experience	Most physicians had a reliance on their past antibiotic prescribing behavior (Mean score = 3.34).	Prior experience is important in physicians’ antibiotic prescribing decision. Physicians were confident in their past prescribing behavior.	Confirmed. Physicians’ personal prescribing experience has a strong influence on antibiotic prescriptions.
Behavioral control	Physicians had good control over antibiotic prescribing (Mean score = 4.00).	Most physicians refused to prescribe antibiotics if unnecessary, even under patient pressure.	Confirmed. Social pressure has limited influence on physicians’ prescribing decision.
Diagnostic tests	Blood routine test was available in all facilities, and 36.7% of facilities could provide C-reactive protein test.	Low utilization of diagnostic tests was common in PHC facilities due to understaffing or patient resistance.	Expanded. Although available, diagnostic tests were not fully utilized in PHC facilities.
Training	Around eighty percent of survey respondents reported that they had received at least one training session on antibiotics in the last twelve months.	Physicians found that these training programs were not very helpful in promoting rational antibiotic use. Additionally, they had limited knowledge of clinical guidelines for antibiotics or acute URTIs.	Expanded. Although most physicians received annual training, the quality of the training was poor and failed to meet PHC physicians’ needs.

## Data Availability

Data is contained within the article or Appendix A.

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
