# Peer review of "Antibiotic Prescribing Decisions for Upper Respiratory Tract Infections Among Primary Healthcare Physicians in China: A Mixed-Methods Approach Based on the Theory of Planned Behavior"

_antibiotics, 2024, doi:10.3390/antibiotics13111104_

Round 1

Reviewer 1 Report

Comments and Suggestions for Authors

Thank you for the opportunity to review this outstanding manuscript describing an innovative convergent mixed-methods study. I agree with your study design, very much like your use of Themes and Subthemes in constructing the questionnaire and interviews, and agree with the way the study results are presented. Your discussion and conclusions are very detailed, relevant, and supported by your study results. I wish you the best with your manuscript.

Reviewer 2 Report

Comments and Suggestions for Authors

This is an interesting study surveying the views of physicians in a certain region of China regarding the treatment of URTIs with or without antibiotics. The study is well designed and the results are well founded and important. A few points to be addressed by the authors:

(a) The authors should mention in the introduction data about the resistance of pathogens involved in URTIs in China (i.e. penicillin resistance of Streptococcus pneumoniae etc)

(b) Are there any data about the vaccination coverage about pathogens involved in URTIs in China? This would also add important info about the epidemiology of URTIs in China. 

(c) Except from hematology tests there are simple tests which can guide the clinician in his/her choice of treatment such as the strep test for Streptococcus pyogenes or the  Paul Bunnel test for infectious mononucleosis. Did the authors investigate the attitude of clinicians in the use of those important, relatively cheap and easy to perform tests?

(d) Are there any data about the kind of prescribed antibiotics? (i.e. b lactams, quinolones etc)

Reviewer 3 Report

Comments and Suggestions for Authors

1-  Line 17 -  I would like to see a reference, or a little explanation on the "Theory of planned  Behavior". 

2- Although the basic topic is not novel, it is an excellent manuscript addressing the practices of prescribing medications for upper respiratory infections; an important topic.

3 - The Figures and Tables are well constructed and presented. I do not have any problems there.   
